# Detection of Filoviruses in Bats in Vietnam

**DOI:** 10.3390/v15091785

**Published:** 2023-08-23

**Authors:** Marat T. Makenov, Lan Anh T. Le, Olga A. Stukolova, Ekaterina V. Radyuk, Evgeny S. Morozkin, Nga T. T. Bui, Olga B. Zhurenkova, Manh N. Dao, Chau V. Nguyen, Mo T. Luong, Dung T. Nguyen, Marina V. Fedorova, Anna V. Valdokhina, Victoria P. Bulanenko, Vasiliy G. Akimkin, Lyudmila S. Karan

**Affiliations:** 1Department of Molecular Diagnostics and Epidemiology, Central Research Institute of Epidemiology, 111123 Moscow, Russia; makenov@cmd.su (M.T.M.); olga.vasilyeva@cmd.su (O.A.S.); radyuk@cmd.su (E.V.R.); zhurenkova@cmd.su (O.B.Z.); culicidae@mail.ru (M.V.F.); valdokhina@cmd.su (A.V.V.); bulanenko@cmd.su (V.P.B.); vgakimkin@yandex.ru (V.G.A.); karan@cmd.su (L.S.K.); 2Biomedicine Institute, Joint Vietnam-Russia Tropical Science and Technology Research Center, Hanoi 122000, Vietnam; leanhbio@gmail.com (L.A.T.L.); pvnhung0506@gmail.com (N.T.T.B.); daonguyenmanh0209@gmail.com (M.N.D.); 3National Institute of Malariology, Parasitology and Entomology, Hanoi 110000, Vietnam; vanchaunimpenk949@gmail.com; 4Southern Branch of Joint Vietnam-Russia Tropical Science and Technology Research Center, Ho Chi Minh City 740500, Vietnam; luongmo@mail.ru (M.T.L.); dungnguyen14791@gmail.com (D.T.N.)

**Keywords:** bat, filovirus, *Dianlovirus*, *Orthomarburgvirus*, *Rousettus leschenaultii*, *Rousettus amplexicaudatus*, Vietnam

## Abstract

A new filovirus named Měnglà virus was found in bats in southern China in 2015. This species has been assigned to the new genus *Dianlovirus* and has only been detected in China. In this article, we report the detection of filoviruses in bats captured in Vietnam. We studied 248 bats of 15 species caught in the provinces of Lai Chau and Son La in northern Vietnam and in the province of Dong Thap in the southern part of the country. Filovirus RNA was found in four *Rousettus leschenaultii* and one *Rousettus amplexicaudatus* from Lai Chau Province. Phylogenetic analysis of the polymerase gene fragment showed that three positive samples belong to *Dianlovirus*, and two samples form a separate clade closer to *Orthomarburgvirus*. An enzyme-linked immunosorbent assay showed that 9% of *Rousettus*, 13% of *Eonycteris*, and 10% of *Cynopterus* bats had antibodies to the glycoprotein of marburgviruses.

## 1. Introduction

The family *Filoviridae* includes eight genera, *Orthomarburgvirus*, *Orthoebolavirus*, *Cuevavirus*, *Oblavirus*, *Striavirus*, *Thamnovirus*, *Tapjovirus*, and *Dianlovirus,* covering 15 species [1]. Four viruses from the family are highly pathogenic to humans and cause severe hemorrhagic fever with a high mortality rate, including *Orthomarburgvirus marburgense* (Marburg virus, MARV), *Orthoebolavirus zairense* (Ebola virus, EBOV), *Orthoebolavirus sudanense* (Sudan virus, SUDV), and *Orthoebolavirus bundibugyoense* (Bundibugyo virus, BDBV). The other viruses of the family are either not highly pathogenic to humans or their pathogenicity has not yet been determined. All filovirus disease outbreaks occurred in Africa, with the exception of imported cases in Europe and the United States [2].

For a long time, it was believed that bats are a natural reservoir of filoviruses [3]. Antibodies to filoviruses were found in 19 bat species in eight countries in Africa, Asia, and Europe [3,4]. In vivo studies have shown that bats are susceptible to filoviruses, replicate the virus, and survive after infection [5,6,7]. Regarding bats trapped in the wild, filoviral RNA was found in 11 bat species in 13 countries. MARV RNA was found in cave-dwelling bats and most frequently in *Rousettus aegyptiacus* [8,9,10,11]. In contrast, despite a number of massive field studies, there is only one publication about successful detection of EBOV RNA in bats [12]. At the same time, for a number of other filoviruses, a relationship between the virus and bats was shown. In particular, Lloviu virus (LLOV) is pathogenic for *Miniopterus schreibersii* and causes epizootics in the colonies of this bat species [13,14]. Bombali virus (BOMV) is most likely associated with the free-tailed bat *Mops condylurus* [15,16,17].

In 2009–2015, new filoviruses were discovered in *Rousettus leschenaultii* and *Eonycteris spelaea* fruit bats caught in Yunnan Province in southern China [18,19]. The whole genome sequence was obtained for only one of the detected filoviruses, which was given the name Měnglà virus (MLAV). Phylogenetic analysis of the coding complete genome sequence indicates that MLAV forms an independent branch with a closer relationship to *Orthomarburgvirus* than *Orthoebolavirus* and *Cuevavirus.* According to the pairwise sequence comparison (PASC) analysis, MLAV fulfils the criteria of a type of virus with a separate genus in *Filoviridae*, and the authors proposed the new genus name *Dianlovirus* [19]. For other filoviruses discovered in Yunnan Province, sequences of sufficient length for taxonomic classification have not been obtained and all of them are designated as unclassified bat filoviruses.

*R. leschenaultii* is widely distributed throughout South and Southeast Asia. Its range extends from Pakistan and India in the west to southern China and Vietnam in the east [20]. The range of *E. spelaea* includes nearly all of Southeast Asia and southern China and extends west through both northwestern and southern Asia [21]. In Vietnam, both bat species mentioned above have a wide but sporadic distribution [22].

In this work, we aimed to find genetic and serological evidence of filovirus infection in bats in Vietnam.

## 2. Materials and Methods

### 2.1. Ethics Statement

The procedures used in this study adhered to the Declaration of Helsinki. Approval was obtained from the ethics committee of the Joint Russia–Vietnam Tropical Science and Technology Research Center (approval number 3225/CN-HDDD).

### 2.2. The Study Areas

Three provinces of Vietnam were selected for fieldwork: Lai Chau and Son La Provinces, located in the north of the country in a mountainous landscape in close proximity to the border with China, and Dong Thap in the south of the country in the Mekong Delta, in a flat area without caves.

### 2.3. Bats Sampling and Processing

The bats were trapped from December 2020 to January 2021 using mist nets (4 m × 20 m). The nets were set in places of potential flight of animals (above streams and in clearings) near caves and abandoned buildings. Standard methods were used for the safe handling and sampling of small mammals that were potentially infected with pathogens [23]. The trapped bats were described morphologically, weighed, and measured (length of the head and body, forearm length, tail, tibia, hind foot, and ear length). Species identification was confirmed for several specimens for each bat species by sequencing the COI gene using the ST-COI-F2 and jgHCO2198 primers [17,24].

Immediately after euthanasia, blood was sampled through cardiac puncture into sterile tubes with 0.5 M EDTA. Oral and rectal swabs were collected in 0.5 mL of 0.15 M NaCl solution. Sections of the brain, liver, spleen, kidney, lung, and intestine were obtained through sterile necropsy. For collection of plasma and cell fractions, blood samples were centrifuged for 5 min at 10,000× *g* at room temperature. All samples obtained were stored in liquid nitrogen during fieldwork and at minus 70 °C in the laboratory.

The brain and intestines were tested separately; internal organs from each animal (liver, spleen, lung, and kidney) were pooled and tested in pools. Bat tissues were homogenized with TissueLyser LT (Qiagen, Hilden, Germany) in 0.5 mL of 0.15 M NaCl solution. Thereafter, total RNA was extracted from 100 µL of 10% suspension by the phenol/chloroform method using a Ribo-Zol kit (AmpliSens, Moscow, Russia) according to the manufacturer’s instructions.

### 2.4. PCR Assays and Sequencing

Isolated RNA was reverse transcribed using a REVERTA-L kit (AmpliSens, Moscow, Russia) according to the manufacturer’s instructions. Amplification of the filovirus polymerase gene fragment was performed with the following primers: FV-F1, FV-R1; FV-F2, FV-R2 [18]; Filo-Mod-FWD, Filo L. conR, Filo-Mod-RVS [15], Marburg F V2 and Marburg R V2 [25] (Appendix A). Polymerase chain reactions (PCR) were performed in a 25 μL volume containing 10 μL PCR buffer master mix (Central Research Institute of Epidemiology, Moscow, Russia), 9 pmol of each primer, and 5 μL of cDNA template. For nested PCR, 1 µL of PCR product from the first PCR round was used in the second PCR round. A 5 μL aliquot of PCR product was electrophoresed on a 1.5% agarose gel and visualized using ultraviolet light after staining with 0.5 μg/mL ethidium bromide. The purified PCR products were sequenced bidirectionally using a BigDye Terminator v1.1 Cycle Sequencing kit (Thermo Fisher Scientific, Austin, TX, USA) on an Applied Biosystems 3500xL Genetic Analyzer (Applied Biosystems, Foster City, CA, USA). The sequences obtained were deposited in NCBI GenBank under the following accession numbers: OP653719–OP653723 (partial RNA-dependent RNA polymerase (RdRp) gene, 309 bp).

### 2.5. Serology Assay

Immunoglobulin G (IgG) antibodies to filoviruses were detected by an enzyme-linked immunosorbent assay (ELISA) using a Human Anti-Marburg (Angola) glycoprotein (GP) IgG ELISA kit (Alpha Diagnostic, San Antonio, TX, USA) and Human Anti-Zaire+Sudan+Reston+Bundibugyo Glycoproteins combo IgG (Alpha Diagnostic International, Inc., San Antonio, TX, USA). The kits were customized for bat IgG detection as follows. Bat plasma was tested at a dilution of 1:250. Goat anti-bat IgG antibodies (Novus Biologicals, Centennial, CO, USA) and peroxidase-labeled rabbit anti-goat IgG antibodies (IMTEK, Moscow, Russia) were used at a dilution of 1:10,000 to detect binding of bat IgG with filovirus antigens sorbed on the bottom of ELISA plates. The following steps of the assay were performed according to the manufacturer’s instructions. Optical density (OD) was measured at 450 nm with baseline correction at 620 nm. Goat anti-bat IgG was preliminarily tested for its ability to bind with IgG of all bat species used during this study.

### 2.6. Data Analysis

The Mega X package was used for phylogenetic analyses. A phylogenetic tree was constructed using the maximum likelihood method with the General Time-Reversible Model [26] and the Hasegawa–Kishino–Yano model with bootstrap analysis based on 1000 replicates.

For customized antifilovirus ELISA cutoff determination, we assumed the presence of a mixture of at least two latent populations for each bat species [27]. We interpreted each latent population in terms of seronegativity and seropositivity as follows: we assumed that the seronegative population was the population with the lowest average value of optical density, while the remaining components were interpreted as different levels of seropositivity. A change point analysis was used to determine the presence of more than one latent population. Furthermore, we extracted the component with the lowest optical density and estimated the model parameters for each of the species with more than one latent population. The 99.9% quantile of the distribution was considered as the cutoff level, which was estimated using parametric bootstrapping. The details of the cutoff calculations are presented in Appendix A. The prevalence and confidence intervals were estimated using exact Clopper-Pearson methods in the package ‘PropCIs’ [28].

## 3. Results

### 3.1. PCR Screening of Filoviruses

A total of 248 bats were studied using three PCR assays. Filovirus RNA was detected only by primers designed by He et al. [18]: FV-F1, FV-R1; FV-F2, FV-R2. The other two PCR assays gave negative results. We found viral RNA in four *R. leschenaultii* and in a single *R. amplexicaudatus* (Table 1). All PCR-positive animals were captured in Lai Chau Province in December 2020. Viral RNA was detected only in pooled internal organs (lung, kidney, liver, and spleen). The samples of the brain tissue, intestine, oral, and rectal swabs were negative.

### 3.2. Genetic Variability of the Filovirus Polymerase Gene Fragment in Studied Bats

A 309 bp region of the RNA-dependent RNA polymerase gene for five PCR-positive samples was sequenced and compared with the available filovirus nucleotide sequences from GenBank. A basic local alignment search tool (BLAST) analysis showed that obtained sequences are most similar to the filoviruses found in China. Phylogenetic analysis revealed that two filovirus sequences Vietnam-R.leschenaultii-119-2020 and Vietnam-R.leschenaultii-122-2020 clustered into a group with the Měnglà virus from *Rousettus* sp. captured in Yunnan Province in China (Figure 1). The third Vietnam-R.amplexicaudatus-91-2020 sequence is identical to the Chinese Bat9447 (GenBank accession number KX371888) and Bat9434 (GenBank accession number KX371883), which form a distinct group closely related to *Dianlovirus* (Figure 1). The sequences of Vietnam-R.leschenaultii-39-2020 and Vietnam-R.leschenaultii-123-2020 form a separate cluster on the phylogenetic tree (Figure 1), which is more closely related to *Orthomarburgvirus* than *Dianlovirus*.

### 3.3. Seroprevalence of Filoviruses in Bats

The change point analysis showed that only the data of *Rousettus*, *Eonycteris*, and *Cynopterus* contained more than one latent population, and as a result, we found IgG-positive animals only for these species. The other bat species had a homogeneous low signal without strong peaks, outliers, or change points, and therefore, we considered them seronegative.

Specific IgG antibodies to filoviruses were detected in 12 fruit bats using the customized Human Anti-Marburg (Angola) glycoprotein (GP) IgG ELISA kit and in 8 fruit bats using the customized Human Anti-Zaire+Sudan+Reston+Bundibugyo Glycoproteins combo IgG kit (Table 1). A total of five MARV IgG-positive samples yielded negative results in ELISA with Ebolavirus antigens, and vice versa, only one IgG-positive sample obtained with Ebolavirus antigens gave negative results with MARV antigen. Consequently, seven samples were IgG-positive by both ELISA kits. Phylogenetic analysis of the glycoprotein gene showed that the filoviruses of the genus *Dianlovirus* are closer to the genus *Orthomarburgvirus* than to *Orthoebolavirus*. Therefore, we used the results of the ELISA with the Marburgvirus antigen for the seroprevalence estimation.

In Lai Chau Province, 9.4% (confidence interval (CI) 95%, 3.1–20.7%) of *R. leschenaultii* bats and 13.3% (CI 95%, 3.8–30.7%) of *E. speleae* bats were seropositive for IgG to filoviruses. The other tested bat species from Lai Chau Province were seronegative, including *Cynopterus* fruit bats. In Son La Province, we trapped only one fruit bat (*R. amplexicaudatus*), and it was seronegative. In contrast, in Dong Thap Province, 10.0% (CI 95%, 1.2–31.7%) of *C. sphinx* fruit bats were seropositive for IgG antibodies against filoviruses (Table 1). We had a plasma sample for only one PCR-positive fruit bat (*R. amplexicaudatus*, id 91), and it was seronegative. In addition to fruit bats, we also tested insectivorous bats of nine species, and all of them were seronegative (Table 1).

## 4. Discussion

Our findings of filoviruses in a northern province of Vietnam extend the previously published evidence of filovirus presence in China. Indeed, we detected filoviral RNA in fruit bats trapped approximately 200 km from the closest sites in Yunnan Province, China, where filoviruses were detected by He et al. and Yang et al. [4,18].

In our work, we studied 248 bats and detected filoviral RNA only with primers designed by He et al. [18]. These primers are located in the most conservative polymerase gene region of known sequences of filoviruses (*Orthomarburgvirus, Orthoebolavirus, Cuevavirus*, and *Dianlovirus* genera). Filoviral RNA was found only in samples from *R. leschenaultii* and *R. amplexicaudatus*. Similar results were obtained by Chinese researchers during their studies in China: they found filovirus RNA only in *Rousettus* sp. and *E*. *spelaea* bats [5,18,19]. The host–pathogen association between filoviruses and bats has not been fully understood to date. However, according to field observations and in vivo experiments, only for MARV we have strong evidence that *R. aegyptiacus* is a reservoir host for the virus [6,10]. There are also data about the association between the LLOV and *M. schreibersii* [13,14], as well as the BOMV and *M. condylurus*, but their role as a reservoir or incidental host is either debatable or requires additional study [3]. Our finding of Asian filoviruses in *Rousettus* fruit bats contributes to the suggestion that *R. leschenaultii* and the closely related *R. amplexicaudatus* are reservoirs for dianloviruses. Additional studies of filovirus infection in these bat species will finally clarify the issue.

We found filovirus RNA only in the internal organs of bats (pools of the lung, liver, spleen, and kidney tissues). The viral RNA was not detected in brain and intestine tissues, oral and rectal swabs for all studied animals. Yang et al. [4] detected dianlovirus RNA in the lung tissues of fruit bats (21 from 21 PCR-positive animals); in a few cases, RNA was detected in other internal organs, i.e., the liver (4 from 21), spleen (3 from 21), and kidney (4 from 21). In comparison with other filoviruses, the MARV was detected in liver, spleen, lung, intestine, kidney, bladder, salivary glands, and female reproductive tract of experimentally infected *R. aegyptiacus* [6], and additionally in lymph nodes, oral swabs, and whole blood of captured, wild *R. aegyptiacus* bats [8]. Ebola virus RNA was found in bat liver and spleen samples [12], and Bombali virus was detected in lung, spleen, liver, heart, intestine, oral swab, and fecal samples of bats [16,17]. Lloviu virus was detected in the lung, liver, rectal swabs, and spleen of bats [13,14]. The tissue tropism of the virus to the different internal organs is important for understanding how the virus is transmitted from one bat to another. Unlike Africa, the lack of data about the detection of filoviruses in oral or rectal swabs in bats studied in Southeast Asia does not allow us to assume the possible ways of filovirus transmission. Additionally, it is important to consider the tissue tropism when planning studies of filoviruses in natural bat populations.

Phylogenetic analysis has shown that three of the five filoviruses found in northern Vietnam belong to the genus *Dianlovirus* and one of them is very close to the Měnglà virus. However, two samples formed a separate clade in an intermediate position between the genera *Dianlovirus* and *Orthomarburgvirus*. We sequenced only a short fragment of the RdRp gene (309 bp), which does not allow us to use the PASC tool [29] to identify the known filovirus species or genus. For a reliable classification of the detected filoviruses, it is necessary to obtain a whole genome sequence. Thus, together with the results obtained in China, our data show that a genetically diverse group of filoviruses circulates in Southeast Asia, represented by several species, possibly belonging to different genera. This assumption is supported in the work of Zhang et al. [30]. Using the ELISA, Western blotting, and neutralization assays, the authors found serological markers of different filovirus species in China.

Our data about the seroprevalence of antibodies to filoviruses in bats in Vietnam showed that the populations of cave-dwelling fruit bats in the Northern Province had been exposed to filoviruses; approximately 9–13% of *Rousettus* sp. and *E. spelaea* had specific antibodies. In the southern province of Vietnam, we caught only *C. sphinx* fruit bats and 10% of them were seropositive to filoviruses. There were no caves in Dong Thap Province, and as a result, there were no cave-dwelling bats (*Rousettus* and *Eonycteris*); therefore, we assume that a filovirus in this province formed a separate focus that consists of forest-dwelling bats (*Cynopterus*) only.

Serological confirmation of bat’s contact with filoviruses was also obtained in China, India, and Singapore. Especially in China, the proportion of seropositive bats reached 56% [4], and the percentage of positive *R. leschenaultii* was 61% [30]. In India, much more modest results were obtained, with 6% of *E. spelaea* and 13% of *R. leschenaultii* being seropositive to filovirus antigens. Serological screening of bat sera samples indicated circulation of several filoviruses [31]. Similar low values of seroprevalence were obtained in Singapore, at 9% of *E. spelaea* [32]. In all mentioned studies, different immunological methods for detecting antibodies and different antigens were used; therefore, these results cannot be directly compared. Our study of seropositivity also had a number of limitations, such as the use of a modified commercial assay and the absence of negative and positive control groups. Subsequent testing for antibodies to filovirus antigens in Western blotting, and in the virus neutralization test, is needed to confirm our initial findings.

The pathogenicity of dianloviruses to humans is not yet known. However, there are data on the detection of antibodies in the blood of people who hunt bats in northeast India [31]. Thus, the detection of antibodies to filoviruses in people who hunt *E. spelaea* and *R. leschenaultii* bats provides evidence for prior exposure of bat hunters to filoviruses. However, cases of acute hemorrhagic fever with Marburg virus-like or Ebola virus-like symptoms have not yet been described in China, India, or northern Vietnam. Apparently, dianloviruses are nonpathogenic to humans or cause mild infection.

## 5. Conclusions

Our finding provides additional evidence that bat filoviruses described in China are more widely distributed than previously suspected. We also have proof that fruit bats of the genus *Rousettus* play a role in the circulation of filoviruses in Asia. Further studies of filoviruses in Asia are needed to prevent the possibility of new, emerging filovirus-related diseases.

## Figures and Tables

**Figure 1 viruses-15-01785-f001:**
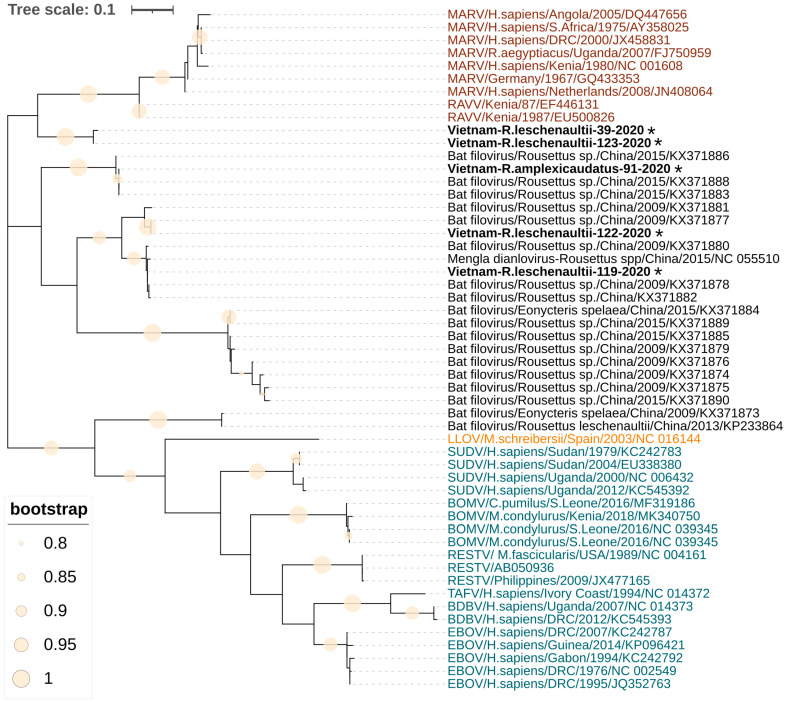
Phylogenetic tree of sequences of filoviruses found in Vietnam. The tree was constructed using the Hasegawa–Kishino–Yano model with the maximum likelihood method (1000 bootstrap replications). A discrete gamma distribution was used to model evolutionary rate differences among the sites. The tree is drawn to scale, with branch lengths corresponding to the number of substitutions per site. Sequences in red font represent the genus *Orthomarburgvirus*, sequences with blue and orange font represent the genera *Orthoebolavirus* and *Cuevavirus*, respectively, and sequences in black represent Dianloviruses and unclassified bat filoviruses. The sequences from fruit bats generated during this study are indicated with asterisks and bold text. The filled circles on branches indicate a bootstrap value greater than 0.8.

**Table 1 viruses-15-01785-t001:** Filovirus detection in bat samples by PCR and customized ELISA kits (Anti-Marburg glycoprotein IgG and Anti-Ebola glycoprotein IgG).

Bat Species	PCR		ELISA		
	Number of Tested Animals	Number of PCR Positive, % (CI, 95%)	Number of Tested Animals	Number of Seropositive with *Marburgviruses* Antigen, % (CI, 95%)	Number of Seropositive with *Ebolaviruses* Antigen, % (CI, 95%)
Lai Chau Province					
*Rousettus leschenaultii*	67	4/6.2 (1.7–15.0)	53	5/9.4 (3.1–20.7)	1/1.9 (0.05–10.10)
*Rousettus amplexicaudatus*	8	1/12.5 (0.3–52.7)	7	1/14.3 (0.4–57.9)	1/14.3 (0.4–57.9)
*Eonycteris spelaea*	35	0/0 *	30	4/13.3 (3.8–30.7)	4/13.3 (3.8–30.7)
*Cynopterus sphinx*	30	0/0	19	0/0	0/0
*Cynopterus horsfieldi*	12	0/0	2	0/0	0/0
*Macroglossus sobrinus*	4	0/0	0	0 *	0 *
*Rhinolophus microglobosus*	16	0/0	21	0/0	0/0
*Hipposideros pomona*	10	0/0	9	0/0	0/0
*Hipposideros cineraceus*	1	0/0	3	0/0	0/0
Son La Province					
*Rousettus amplexicaudatus*	1	0/0	1	0/0	0/0
*Rhinolophus chaseni*	1	0/0	2	0/0	0/0
*Rhinolophus pusillus*	23	0/0	21	0/0	0/0
*Taphozous melanopogon*	3	0/0	2	0/0	0/0
Dong Thap Province					
*Cynopterus sphinx*	0	not applicable	20	2/10.0 (1.2–31.7)	2/10.0 (1.2–31.7)
*Myotis hasseltii*	25	0/0	22	0/0	0/0
*Taphozous longimanus*	12	0/0	12	0/0	0/0
*Scotophilus kuhlii*	0	not applicable	3	0/0	0/0

*—confidence intervals were not calculated for zero signal.

## Data Availability

The data presented in this study are available on request from the corresponding author.

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
