# Peer review of "Detection of Filoviruses in Bats in Vietnam"

_viruses, 2023, doi:10.3390/v15091785_

Round 1
Reviewer 1 Report
The article "Detection of filoviruses in bats in Vietnam" describes the presence of filoviruses in Vietnam and presents genomic and serological evidence. The article describes novel Dianlo- and Marburg-associated sequences. The paper is clear and well written. Sequencing of the GP gene would have provided additional information on the exact positioning of the described viruses within the filoviruses. There is no mention that RT-PCR analyses were performed on the animals' sera, even though they were sampled for serology. This point is of particular interest as the presence of virus in sera (as described for Lloviu, Bombali) could help to minimise the culling of animals.
Reviewer 2 Report
this study is about detecting filovirus genome and seroconversion in bats in Vietnam. the study is of interest and support the ideas of the presence of filoviruses in bats in Asia.
I made some comments that could benefit the manuscript:
-it is unclear whether some subjects positive for MARV GP antibodies were also positive for viral RNA.
-why not all the bats were tested for seroconversion?
-Please clarify in the M&M whether FV primers (the only ones that worked here) are more relevant to detect MARV-like versus EBOV-like viruses. Is the 300bp of the polymerase a conserved region for both Orthoebolavirus and Orthomarburgvirus? Readers will want to know that.
-line 34 there are only 4 highly pathogenic species. TAFV did not cause death in the single reported human case.
-Justify why MARV GP was chosen as antigen instead of the common EBOV GP. Justify the 1:250 dilution of all sera.
-Table 1 seem to be missing
-comment on the cutoff calculation for ELISA: the calibrator you used is the lowest provided in the kit but it does not not necessarily mean this is the lowest amount of antibody you could have detected above background. how far off the cut-off was between the method A of your kit (threshold index determination) and after change point analysis? Also if the 99.9% quantile of the distribution of the dataset with the lowest OD was chosen to determine the cut-off value why is that value so far from the change point (in dataset 1)?
Minor editing of English language required
